# A systematic review of evidence on employment transitions and weight change by gender in ageing populations

Alexander C. T. Tam[1]*, Veronica A. Steck[2], Sahib Janjua[3], Ting Yu Liu[3], Rachel A. Murphy[1,4], Wei Zhang[1,5‡], Annalijn I. Conklin[5,6‡]

1 Faculty of Medicine, School of Population and Public Health, University of British Columbia, Vancouver, British Columbia, Canada, 2 Faculty of Science, Department of Life Sciences, McGill University, Montreal, Quebec, Canada, 3 Faculty of Pharmaceutical Sciences, University of British Columbia, Vancouver, British Columbia, Canada, 4 Cancer Control Research, BC Cancer Research Institute, Vancouver, British Columbia, Canada, 5 Centre for Health Evaluation and Outcome Sciences (CHÉOS), Providence Research, Vancouver, British Columbia, Canada, 6 Faculty of Pharmaceutical Sciences, Collaboration for Outcomes Research and Evaluation (CORE), University of British Columbia, Vancouver, British Columbia, Canada

‡ These authors contributed equally to this work and are joint senior authors.
* atam96@student.ubc.ca

**Data Availability Statement:** All relevant data are within the article and its Supporting Information files.

## Abstract

### Background

Becoming unemployed is associated with poorer health, including weight gain. Middle- and older-age adults are a growing segment of workforces globally, but they are also more vulnerable to changes to employment status, especially during economic shocks. Expected workforce exits over the next decade may exacerbate both the obesity epidemic and the economic burden of obesity. This review extends current knowledge on economic correlates of health to assess whether employment transitions impact body weight by sex/gender among middle-aged and older adults.

### Methods

Eight bibliometric databases were searched between June and July 2021, supplemented by hand-searches, with no restriction on publication date or country. Longitudinal studies, or reviews, were eligible when examining body weight as a function of employment status change in adults ≥50 years. Data extraction and quality appraisal used predefined criteria; reported findings were analysed by narrative synthesis.

### Results

We screened 6,001 unique abstracts and identified 12 articles that met inclusion criteria. All studies examined retirement; of which two also examined job-loss. Overall, studies showed that retirement led to weight gain or no difference in weight change compared to non-retirees; however, reported effects were not consistent for either women or men across studies or for both women and men within a study. Reported effects also differed by occupation: weight gain was more commonly observed among retirees from physical occupations but

**Funding:** The authors received no specific funding for this work.

**Competing interests:** The authors have declared that no competing interests exist.

not among retirees from sedentary occupations. Few studies assessed the role of health behaviours; sleep was the least studied. Most studies were medium quality.

## Conclusions

Existing studies do not provide a clear enough picture of how employment transitions affect body weight. Firm conclusions on the impact of employment transitions on weight cannot be made without further high-quality evidence that considers the role of gender, job-type, other health behaviours, and other transitions, like job-loss.

## Introduction

Work confers to individuals a sense of identity and coherence [1], social networks [2], and a stream of income that provides stability in meeting one's material needs [2,3]. It also represents a significant amount of time spent away from other activities. In older adults, work also serves as a way of staying socially, mentally, and physically active and to continue contributing to the community and the broader economy [3,4]. Across OECD countries, the average share of the workforce aged 45 to 59 is expected to grow from 26% (1995) to 34% by 2030, and the share of workers aged 60 and above is projected to increase from 4% to 9% [5]. Coupled with the rise in labour force participation of middle-aged and older workers are unique challenges such as lower rates of enrolment into continued education and skills training [5], employer preference for hiring younger employees [5], and ageism at work [5,6]. Thus, more middle-aged and older workers may find themselves experiencing job-loss and challenges with re-employment following economic shocks, as observed following the Great Recession in the late 2000s and currently with the COVID-19 pandemic [7,8].

Beyond the direct financial and economic consequences of employment transitions, middle-aged and older workers who experience later-life career loss are also likely to experience worsening health [9]. In particular, involuntary job-loss in middle-aged and older adults is associated with declines in self-reported physical functioning [9], and increased risk of stroke and cardiovascular disease [10,11]. One of the risk factors for cardiovascular disease that has received some attention in the employment literature is body weight change [12–16], and how it differs following different changes in employment status. Changes in employment status, or employment transitions, span a range of events from exiting the labour force through job-loss and retirement to re-entry into the labour force by re-employment. Loss of a job without compensation has been found to be associated with both decreases and increases in body weight; [12,15] while voluntary and planned retirement has been found to be associated with weight gain [16]. However, these results have not been consistently supported in the literature [17,18]. It has been noted that change in body weight may depend on gender [12], occupation type [14], and changes in physical activity levels following a transition [14].

One additional post-transition behaviour change that may also be important in the effect of employment transitions on body weight is sleep [19–25]. For older adults (age 65 and older), the National Sleep Foundation recommends between 7 and 8 hours of sleep per night [26]. Sleep durations that are shorter and longer than recommended ranges are associated with increased risk of incident obesity, cardiovascular disease, type 2 diabetes, and all-cause mortality [27].

In a cohort of public employees in the U.S. [19], individuals who transitioned from full-time employment to full retirement had bedtimes and wake times that were 30 minutes and 63 minutes later in their first year post-retirement, respectively [19]. These delays increased to 36

and 78 minutes respectively by the third year [19]. The later bedtime and much later wake time led to a significant increase in overall sleep duration during weekday nights after retirement [19]. This increase in overall sleep duration may also be coupled with improvements in sleep quality [25] which may be protective rather than harmful, especially among those who were at higher risk of sleep disturbances due to job demand, fatigue, and poor mental health initially [25]. Additionally, in a study of Finnish public employees [20], a decline in sleep difficulties such as non-restorative sleep and waking up too early was observed in the period immediately following retirement [20]. Aside from retirement, the experience of job-loss is also associated with differential sleep outcomes. In one longitudinal study of aggregated employment data in the U.S. [28], those who experience periods of unemployment were more likely to have shorter and longer sleep durations compared to those who were employed continuously. Those who experienced unemployment were also more likely to report sleep disturbances and a greater number of nights of insufficient sleep [28].

In turn, poor sleep quality and both too much and too little sleep have been associated with increases in body weight in older adults. A literature review published in 2018 reviewed the existing evidence of sleep parameters in association with obesity in older adults [29]. Included longitudinal studies suggested that both shorter and longer sleep durations were associated with increased risk of weight gain,[30–32] and increased odds of obesity compared to those with recommended sleep durations in the order of 2.3-fold among women and 3.7-fold among men [33].

In sum, there is a potential mediating role that sleep may play in the impact of employment transitions on subsequent body weight gain. Sleep may contribute to explaining the heterogeneity that appears to exist in the literature on job-loss and retirement and body weight gain in older adults, yet it has not received as much attention as physical activity [14,18], or other modifiable health behaviours (e.g., alcohol consumption and smoking) [15,16]. To address this gap, this review has two objectives: (1) to systematically search the existing literature on the impact that employment transitions have on body weight by women and men among middle-aged and older adults, and (2) to examine whether sleep confounds, mediates, or modifies the effect that employment transitions have on body weight in the identified studies.

## Methods

### Search and data sources

Journal articles were systematically searched using eight bibliometric databases (Ovid MEDLINE/PubMed, Scopus, PsycINFO, Web of Science, Embase, EconLit, CINAHL, and Applied Social Sciences Index and Abstracts). Table 1 contains the search terms used in the databases after consulting with a reference librarian. Free-text thesaurus and MeSH terms were used with Boolean operators to capture the concepts of "middle-age and older", "employment", "transition", and "body weight". No restrictions were placed on publication date, country, or original publication language. Screening, assessment, and inclusion were limited to studies published in English, French, and Chinese. Searches were performed by ACTT, VAS, SJ, and TYL between June and July 2021. Search strategies in each of the databases are described in supplementary material (S1 Table). This systematic review was not registered.

### Screening

Two reviewers (ACTT and VAS, SJ, or TYL) screened titles and abstracts independently. Records were removed based on exclusion criteria and eligible full-texts were retrieved and screened for inclusion and reference-tracing adhering to the same division of work between

**Table 1. Search terms used in literature search.**

| Concept | Search terms ("/" indicating "OR") |
|---|---|
| **Middle-age and older adults** | old* adult*/ ag?ing / aged / elder* / geriatric* / senior |
| **Employment** | employ* / job* / unemploy* / work* / retir* |
| **Transition** | change / loss / transition / terminat* / dismiss* / lay-off / reduc* / becom* / enter* / adjust* |
| **Weight** | weight / bmi / body mass index / adipos* / "met* syndrome" / "cardiovascular disease" |

* indicates a search of the provided root and any ending.

? indicates a wildcard replacement of zero or one character. Variation in wildcard symbols were accounted for and the appropriate alternative symbols were used (e.g., $ or #) on a per database basis.

two independent reviewers. All disagreements were managed with discussion among reviewers and resolved by consensus.

## Criteria

A study was eligible for inclusion if it examined the longitudinal association of employment transition (e.g., a change in employment status between two time-points) and subsequent body weight outcomes in adults aged 50 years of age and older. Studies were excluded if the exposure of interest was employment status at only single point in time as the objective of this review is to examine how changes to employment status may be associated with body weight outcomes. Examples of changes in employment status are job-loss (employed at baseline of study and unemployed at follow-up) and retirement (employed at baseline of study and retired and not working at follow-up). No restrictions were placed on how studies defined the transitions or body weight outcomes. Additional exclusion criteria include: cross-sectional design; qualitative studies; broad age groups with no stratification of results (e.g., 18 years to 65 years); non-body weight outcomes; and, clinical populations. Studies that also included an analysis with sleep parameters were considered for our second objective; however, the absence of sleep as a variable in the analysis was not a criterion for exclusion. A summary of the criteria used are in Table 2.

## Risk of bias assessment, data extraction, and analysis

Studies were appraised using the Effective Public Health Practice Project (EPHPP) tool given its suitability for assessing quantitative studies based on observational data [34]. The EPHPP

**Table 2. Inclusion/exclusion criteria used.**

| | Inclusion criteria | Exclusion criteria |
|---|---|---|
| Population | Adults in their middle-age or older | Adults in early adulthood; broad age groups, unless results are stratified; clinical populations |
| Exposure | Change in employment status | Only static employment status (e.g., baseline employment as exposure) |
| Outcome(s) | Change in self-reported or objectively measured body weight | None |
| Types of studies | Longitudinal studies (RCTs, cohort, panel/ecological, case-control) | Cross-sectional studies, qualitative studies, editorials, and validation studies |
| Setting | Any settings | None |
| Publication year | Any year | None |
| Publication language | English, French, and Chinese | Not published in English, French, and Chinese |

quality assessment tool contains 21-items with sections that cover components such as selection bias, study design, confounding, data collection, and analysis considerations. Of the 21 items, 11 are directly relevant in assigning a global rating of "strong", "moderate", or "weak" to our included studies. The remaining 10 items from the tool are not relevant to the studies that were included in our systematic review and thus, had no bearing on the scoring assignment. For example, the items include questions related to randomized controlled trial design, the integrity of the intervention received, and intention-to-treat analytic approaches [34]. ACTT and VAS independently assessed study quality and initial assessments were compared, with final ratings established by consensus.

VAS, SJ, and TYL extracted data from studies into a standardised evidence table: author; objectives; study time frame; setting; research design; sample characteristics; description of exposure; description of outcome; time to follow-up; job-types specified; and, body weight results. Where relevant, gender and job-type stratified results are also populated into the results field. ACTT reviewed the extracted data and provided feedback to the three extractors during the process. A narrative synthesis was used to summarize the findings. The narrative synthesis revealed additional health behaviours beyond sleep that were studied as mechanisms; thus, a post-hoc analysis was conducted to summarise additional results.

## Results

The results of the database searches are presented in a flow diagram in Fig 1. Database searches and citation tracing identified 6,001 unique records after duplicates were removed (n = 1,754).

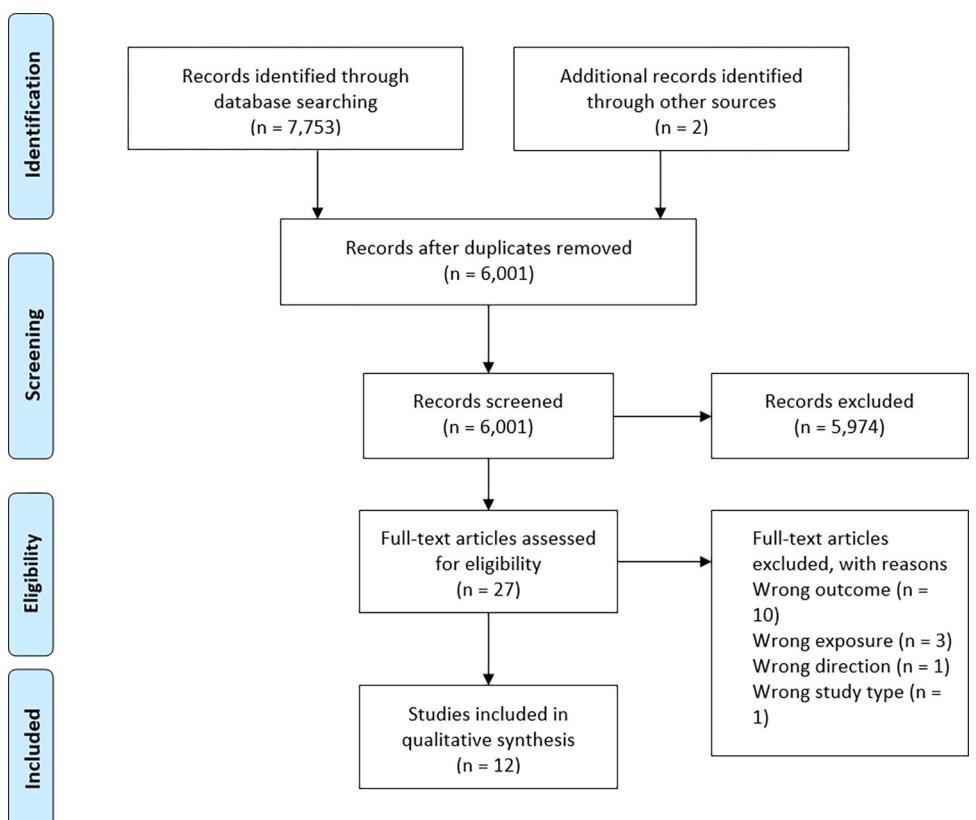

**Fig 1. Modified PRISMA flow diagram of literature search and study selection** [35].

In total, 27 original research articles remained after title and abstract screening and were read in full. Of the 27, 12 were eligible for inclusion and were assessed for risk of bias, extracted from, and synthesized.

Fifteen studies were excluded after full-text review. Ten studies were excluded because results were not relevant to the outcomes of interest (body weight) [36–45]. Three studies used employment status at a single time point and thus, did not measure the exposure of interest (employment *transition*) [46–48]. The two remaining studies investigated body weight as a determinant of retirement decisions and did not have results relevant to the desired direction of association [49]; or was a protocol document [50].

## Characteristics of included studies

The characteristics of the included studies are presented in Table 3. All 12 included studies analyzed the transition from employment into retirement, while only two additionally

**Table 3. Characteristics of included studies.**

| Author / source | Stated study objective | Years | Setting | Study design, duration (data) | Study population (n) | Description of exposure | Outcome measures |
|---|---|---|---|---|---|---|---|
| Morris et al. 1992 [16] [ProQuest] | To assess the effect of unemployment and early retirement on cigarette smoking, alcohol consumption, and body weight in a group of middle-aged British men | 1978–1980; 1983–1985 | UK, primary care | Prospective cohort 5 y follow-up (British Regional Heart Study) | Men (40–59 y) (n = 6,191) | Baseline employment status and reasons for subsequent change were used to categorize men into transitions: (1) **Continuously employed** (2) **Discontinuously employed** (3) **Unemployed due to illness** (4) **Retired due to illness** (5) **Unemployed not due to illness** (6) **Retired not due to illness** | Anthropometric measures: weight (Kg), height (cm) Self-reported body mass index (BMI) *% change* in body weight Smoking (never, light, moderate, heavy), alcohol consumption (non-drinker, occasional, light, moderate, heavy), physical activity (PA) (regular, recreational, sporting) |
| Nooyens et al. 2005 [18] [MEDLINE] | To evaluate the impact of retirement on diet, physical activity, body mass index (BMI) and waist circumference over a 5-year follow-up period in a population-based cohort | 1987–1991; 1993–1997; 1998–2002 | Town of Doetinchem, The Netherlands, at the municipal health centre | Prospective cohort 5 y follow-up | Men (50–65 y) (n = 288) | **Retired:** respondent indicated they are retired at follow-up **Continue working:** respondent indicated they are still working at follow-up | Anthropometric measures: weight (Kg), height (cm), waist circumference (cm) Leisure-time PA (hrs/wk from EPIC physical activity questionnaire), diet (portion and frequency of items from EPIC Food Frequency Questionnaire) |
| Forman-Hoffman et al. 2008 [17] [MEDLINE] | To determine whether retirement is associated with either weight gain or weight loss | 1994–2002 | USA | Prospective cohort 2 y follow-up (Health & Retirement Study) | Adults (53–63 y) (n = 10,150) | **Continuing to work:** respondents reported working at least 20 hrs/month in adjacent interviews (e.g. 1994 and 1996) **Recently retired:** respondents reported currently working at the time of the first interview but reported retiring in the previous 2 years at the time of the next biennial interview | *% change* in self-reported BMI Vigorous PA (started, stopped, no change) |
| Zheng 2008 [14] [ProQuest] | To investigate the long-term effects of physical activity on body weight and the effect of food price on body weight | 1992–2006 | USA | Prospective cohort 2 y follow-up (Health & Retirement Study) | Men (51–61 y) Women (50–61 y) (n = 3,936) | **Retired:** respondents indicated they are retired as opposed to working or unemployed **Not retired:** respondents indicated they were working or unemployed | Self-reported BMI Vigorous PA (≥3 times/week, <3 times/week) |

(*Continued*)

**Table 3.** (Continued)

| Author / source | Stated study objective | Years | Setting | Study design, duration (data) | Study population (n) | Description of exposure | Outcome measures |
|---|---|---|---|---|---|---|---|
| Chung et al. 2009 [52] [MEDLINE] | To investigate the effect of retirement on the change in body mass index (BMI) in diverse groups varying by wealth status and occupation type | 1992–2002 | USA | Prospective cohort 2 y follow-up (Health & Retirement Study) | Adults (50–71 y) (n = 10,565) | **Retired:** respondents who at the time of the interview were not working for pay **Currently working:** respondents who at the time of the interview were working for pay | Self-reported BMI |
| Gueorguieva et al. 2011 [55] [Hand search] | To determine whether retirement influences BMI patterns according to occupational affiliation | 1992–2002 | USA | Prospective cohort 2 y follow-up (Health & Retirement Study) | Adults (50+ y) (n = 2,096) | **Retired:** respondents indicated they were working during their initial interview and subsequently indicated they had retired during follow-up surveys | Self-reported BMI Annualized *change* in BMI |
| Monsivais et al. 2015 [12] [MEDLINE] | To examine the association between employment change and weight change among three groups: Those who maintained employment over the follow-up period, those who retired and those who became unemployed | British Household Panel Survey: 2004–2005; 2006–2007 EPIC-Norfolk: 1993–1997; 1998–2000 | UK | Longitudinal 2.2 y follow-up (British Household Panel Survey) 3.5 y follow-up (EPIC-Norfolk cohort) | Adults (18+ y), BHPS (n = 4,539) Adults (39–75 y), EPIC-Norfolk (n = 7,201) | **Remained employed:** BHPS: respondent reporting being employed at follow-up EPIC-Norfolk: respondent reported working at follow-up **Retired:** BHPS: respondent reported being retired at follow-up EPIC-Norfolk: respondent reported being retired and either working or not working at follow-up **Lost job:** BHPS: respondent reported being unemployed at follow-up EPIC-Norfolk: respondent reported being unemployed; unemployed and retired; not retired or unemployed but out of work for other reasons | Anthropometry: weight (Kg), height (cm) (EPIC-Norfolk) Annualized *change* in body weight Sleep loss due to worry (4 categories to GHQ12: "Have you recently lost sleep over worry?") |
| Godard 2016 [56] [Hand search] | To estimate the causal impact of retirement on BMI, on the probability of overweight or obesity and on the probability of obesity | 2004; 2006; 2010 | Europe (Austria, Spain, Germany, Italy, Sweden, France, Switzerland and Belgium) | Panel 2 y follow-up 4 y follow-up | Employed adults (50–69 y) (n = 2,599) | **Retired:** respondents indicated they were currently retired and were previously currently working **Continuously employed:** respondents indicated they were currently working for all waves of data | Self-reported BMI *Change* in obesity (kg/m$^2$) (underweight, normal, overweight, obese) Leisure-time PA (4 categories to "How often do you engage in activities that require a moderate level of energy, such as gardening, cleaning the car, or going for walk?") |
| Stenholm et al. 2017 [53] [MEDLINE] | To examine changes in body mass index during years preceding retirement, during retirement transition and after retirement | Pre-retirement: 2000–2002; 2004; 2008 Post-retirement: 2005, 2009, 2013 | Ten towns (Tampere, Espoo, Turku, Vantaa, Oulu, Raisio, Naantali, Valkeakoski, Nokia, and Virrat), Finland | Prospective cohort 4 y follow-up | Retired adults (late 50 y) (n = 5,426) | **Statutory retirement:** respondents indicated they fully retired at the statutory retirement age and did not report retirement due to health grounds, unemployment, or partial retirement plans | Self-reported BMI *Change* in obesity (kg/m$^2$) (underweight, normal, overweight, obese) |
| Syse et al. 2017 [57] [PsycINFO] | To assess the association between retirement and 5-year changes in health and health behaviour | 2002–2007 | Norway | Panel study 5 y follow-up | Employed adults (57–66 y) (n = 546) | **Retired:** respondents who were collecting pension at T2 (2007) were considered retired voluntarily or involuntarily **Still working:** respondents who reported still working at T2 | *Change* in 1 Kg (self-reported) |

(*Continued*)

**Table 3.** (Continued)

| Author / source | Stated study objective | Years | Setting | Study design, duration (data) | Study population (n) | Description of exposure | Outcome measures |
|---|---|---|---|---|---|---|---|
| Feng et al. 2020 [54] [MEDLINE] | To investigate the effect of retirement on body mass index and body weight | 2011–2015 | China | Panel study 2 y follow-up (China Health and Retirement Survey) | Retired adults (45 + y) (n = 3,090) | **Retired**: respondents indicated they are completely retired, do not work any hours in the labour market, and held a paid job at one point in the past **Non-retirees**: respondents indicated they work in the labour market and are not retired | BMI from measured weight (Kg), height (cm) Leisure-time PA (yes, no to "during a usual week, did you do any physical exercise for at least ten minutes continuously."), diet (food consumption and spending per capita) |
| Pedron et al. 2020 [51] [MEDLINE] | To estimate the causal effect of retirement on a large set of biomedical and behavioral risk factors for cardiovascular and metabolic disease | Baseline: 1994–1995; 1999–2000 Follow-up: 2004–2005; 2006–2008; 2013–2014 | Germany | Panel study 6–10 y follow-up (Cooperative Health Research in the Augsburg Region study) | Adults (45 –80 y) (n = 11,168) | **Retired**: respondents reported their current employment status as "retired" **Not retired**: respondents indicated any status that was not "retired" such as employed, unemployed, homemakers, and unemployed due to long-term sickness | BMI from measured weight (Kg), height (cm) |

considered transitions from employment into unemployment (job-loss) [12,16]. Analytic samples ranged in sizes from 288 to 11,168 respondents [18,51]. Eight of the studies used change in body mass index (BMI) between time points as their body weight outcome [14,16,51–56]. Additional outcomes included odds of weight change [17,57], odds or risk of overweight or obesity [53,56], and annualized weight change [12]. Measured body size data was used in five studies [12,16,18,51,54], while the other seven used self-reported measures [14,17,52,53,55–57]. Over half (n = 7) of the included studies utilized cohort data [14,16–18,52,53,55], while four used panel data [51,54,56,57], and one study used both cohort and panel data [12]. Time to follow-up also varied from 2 years [56] to 10 years [51]. Two studies restricted their study samples to men [16,18], and three studies did not report results by sex/gender [52,55,57].

## Quality of included studies

Two studies were rated as weak [53,54]; a majority of the studies (n = 8) were considered moderate in quality [12,14,16–18,51,55,57]; and two were rated as strong [52,56] (Table 4). The reasons for the weak rating included an absence of description and count of those lost-to-follow-up or drop-outs in the data source, as well as a lack of discussion on the validity or biases of the relevant data collection tools. All studies controlled for many relevant confounders through design (stratification of sample by health status or initial body weight class) or through statistical adjustment in models, while only three studies described the validity of using self-reported body weight in their respective populations [17,52,56].

Table 5 provides a summarised overview of data extracted from included studies with respect to the direction of change in the body weight outcome by employment transition, gender, occupation type, and comparator (pre-post or against a control). Detailed extraction of included study results and definitions of occupation types are provided as online supporting information (S2 and S3 Tables).

**Table 4. Quality appraisal of included studies.**

| Section | Question number | Question | Monsivais et al. 2015 [12] | Morris, Cook, and Shaper 1992 [16] | Gueorguieva et al. 2011 [55] | Godard 2016 [56] | Nooyens et al. 2005 [18] | Pedron et al. 2020 [51] | Forman-Hoffman et al. 2008 [17] | Feng et al. 2020 [54] | Chung et al. 2009 [52] | Zheng 2008 [14] | Syse et al. 2017 [57] | Stenholm et al. 2017 [53] |
|---|---|---|---|---|---|---|---|---|---|---|---|---|---|---|
| A | Q1 | Are the individuals selected to participate in the study likely to be representative of the target population? | 1—Very likely | 1—Very likely | 1—Very likely | 2—Somewhat likely | 2—Somewhat likely | 1—Very likely | 1—Very likely | 1—Very likely | 1—Very likely | 1—Very likely | 2—Somewhat likely | 1—Very likely |
| A | Q2 | What percentage of selected individuals agreed to participate? | 5 Can't tell | 2 60–79% agreement | 5 Can't tell | 2 60–79% agreement | 2 60–79% agreement | 5 Can't tell | 1 80–100% agreement | 5 Can't tell | 1 80–100% agreement | 5 Can't tell | 2 60–79% agreement | 5 Can't tell |
| A | Rate this section | 1—strong 2—moderate 3—weak | 1—strong | 2—moderate | 1—strong | 2—moderate | 2—moderate | 1—strong | 1—strong | 1—strong | 1—strong | 1—strong | 2—moderate | 1—strong |
| B | Q1 | STUDY DESIGN Indicate the study design | 3—Cohort Analytic | 3—Cohort Analytic | 3—Cohort Analytic | 7—Panel | 3—Cohort Analytic | 7—Panel | 3—Cohort Analytic | 7—Panel | 3—Cohort Analytic | 5—Cohort | 7—Panel | 5—Cohort |
| B | Rate this section | 1—strong 2—moderate 3—weak | 2—moderate | 2—moderate | 2—moderate | 2—moderate | 2—moderate | 2—moderate | 2—moderate | 2—moderate | 2—moderate | 2—moderate | 2—moderate | 2—moderate |
| C | Q1 | Were there important differences between groups prior to the intervention? | 1—Yes | 1—Yes | 1—Yes | 1—Yes | 1—Yes | 1—Yes | 1—Yes | 1—Yes | 1—Yes | 1—Yes | 1—Yes | 1—Yes |
| C | Q2 | If yes, indicate the percentage of relevant confounders that were controlled (either in the design (e.g. stratification, matching) or analysis)? | 1—80–100% (most) | 2—60–79% (some) | 2—60–79% (some) | 1—80–100% (most) | 2—60–79% (some) | 2—60–79% (some) | 1—80–100% (most) | 1—80–100% (most) | 1—80–100% (most) | 2—60–79% (some) | 1—80–100% (most) | 2—60–79% (some) |
| C | Rate this section | 1—strong 2—moderate 3—weak | 1—strong | 2—moderate | 2—moderate | 1—strong | 2—moderate | 2—moderate | 1—strong | 1—strong | 1—strong | 2—moderate | 1—strong | 2—moderate |

(*Continued*)

**Table 4.** (Continued)

| Section | Question number | Question | Stenholm et al. 2017 [53] | Syse et al. 2017 [57] | Zheng 2008 [14] | Chung et al. 2009 [52] | Feng et al. 2020 [54] | Forman-Hoffman et al. 2008 [17] | Pedron et al. 2020 [51] | Nooyens et al. 2005 [18] | Godard 2016 [56] | Gueorguieva et al. 2011 [55] | Morris, Cook, and Shaper 1992 [16] | Monsivais et al. 2015 [12] |
|---|---|---|---|---|---|---|---|---|---|---|---|---|---|---|
| D | Q1 | Was (were) the outcome assessor(s) aware of the intervention or exposure status of participants? | 1—Yes | 1—Yes | 1—Yes | 1—Yes | 1—Yes | 1—Yes | 1—Yes | 1—Yes | 1—Yes | 1—Yes | 1—Yes | 1—Yes |
| D | Q2 | Were the study participants aware of the research question? | 2—No | 2—No | 2—No | 2—No | 2—No | 2—No | 2—No | 2—No | 2—No | 2—No | 2—No | 2—No |
| D | Rate this section | 1—strong 2—moderate 3—weak | 2—moderate | 2—moderate | 2—moderate | 2—moderate | 2—moderate | 2—moderate | 2—moderate | 2—moderate | 2—moderate | 2—moderate | 2—moderate | 2—moderate |
| E | Q1 | Were data collection tools shown to be valid? | 3—Can't tell | 3—Can't tell | 3—Can't tell | 1- Yes | 3—Can't tell | 1- Yes | 3—Can't tell | 3—Can't tell | 1- Yes | 3—Can't tell | 3—Can't tell | 3—Can't tell |
| E | Q2 | Were data collection tools shown to be reliable? | 3—Can't tell | 3—Can't tell | 3—Can't tell | 3—Can't tell | 3—Can't tell | 3—Can't tell | 3—Can't tell | 3—Can't tell | 3—Can't tell | 3—Can't tell | 3—Can't tell | 3—Can't tell |
| E | Rate this section | 1—strong 2—moderate 3—weak | 3—Weak | 3—Weak | 3—Weak | 2—Moderate | 3—Weak | 2—Moderate | 3—Weak | 3—Weak | 2—Moderate | 3—Weak | 3—Weak | 3—Weak |
| F | Q1 | Were withdrawals and drop-outs reported in terms of numbers and/or reasons per group? | 2—No | 1—Yes | 1—Yes | 1—Yes | 3—Can't tell | 3—Can't tell | 1—Yes | 1—Yes | 1—Yes | 1—Yes | 1—Yes | 1—Yes |
| F | Q2 | Indicate the percentage of participants completing the study. (If the percentage differs by groups, record the lowest). | 4—Can't tell | 2—60–79% | 1–80–100% | 1–80–100% | 4—Can't tell | 4—Can't tell | 2–60–79% | 2–60–79% | 2–60–79% | 1–80–100% | 1–80–100% | 2–60–79% |
| F | Rate this section | 1—strong 2—moderate 3—weak Not applicable | 3—Weak | 2—Moderate | 1—Strong | 1—Strong | 3—Weak | 3—Weak | 2—Moderate | 2—Moderate | 2—Moderate | 1—Strong | 1—Strong | 2—Moderate |

(Continued)

**Table 4.** (Continued)

| Section | Question number | Question | Stenholm et al. 2017 [53] | Syse et al. 2017 [57] | Zheng 2008 [14] | Chung et al. 2009 [52] | Feng et al. 2020 [54] | Forman-Hoffman et al. 2008 [17] | Pedron et al. 2020 [51] | Nooyens et al. 2005 [18] | Godard 2016 [56] | Gueorguieva et al. 2011 [55] | Morris, Cook, and Shaper 1992 [16] | Monsivais et al. 2015 [12] |
|---|---|---|---|---|---|---|---|---|---|---|---|---|---|---|
| Global Rating | | | 3—Weak | 2—Moderate | 2—Moderate | 1—Strong | 3—Weak | 2—Moderate | 2—Moderate | 2—Moderate | 1—Strong | 2—Moderate | 2—Moderate | 2—Moderate |

*Note.* The global rating is based on the number of "Weak" ratings across the six sections. An article is given a global "Weak" rating if 2 or more sections receive a "Weak" rating; a "Moderate" rating is given if only one section was considered "Weak"; and, a "Strong" paper has no section "Weak" ratings. The Effective Public Health Practice Project tool contains two additional sections that do not contribute to the global rating so the scores are omitted from this table.

**Table 5. Summarised results from included studies by study design, employment transition, and occupation type.**

| | Measured anthropometry | | | | Self-reported anthropometry | | | | | | | |
|---|---|---|---|---|---|---|---|---|---|---|---|---|
| | Nooyens et al [18] | Monsivais et al [12] | Feng et al [54] | Pedron et al [51] | Morris et al [16] | Forman-Hoffman et al [17] | Zheng [14] | Chung et al [52] | Gueorguieva et al [55] | Godard [56] | Stenholm et al [53] | Syse et al [57] |
| **Weight change relative to not retired/continued to work** | | | | | | | | | | | | |
| *Retirement* | | | | | | | | | | | | |
| Active | / | / | / | / | / | W + <br> M + | W + <br> **M +** | **A +** | / | W + <br> **M +** | / | / |
| Sedentary | / | / | / | / | / | W + - <br> M - | W + <br> **M -** | A + | / | W + <br> M + | / | / |
| Overall | / | W + <br> M - | W + - <br> **M +** | **W +** <br> M + | M + | W + <br> M + - | / | **A +** | / | **W +** <br> **M +** | / | **A -** |
| *Lost job* | | | | | | | | | | | | |
| Active | / | / | / | / | / | / | / | / | / | / | / | / |
| Sedentary | / | / | / | / | / | / | / | / | / | / | / | / |
| Overall | / | **W +** <br> M + | / | / | **M +** | / | / | / | / | / | / | / |
| **Weight change within employment transition (pre-post change)** | | | | | | | | | | | | |
| *Retirement* | | | | | | | | | | | | |
| Active | M + | / | / | / | / | / | / | / | *A +* | / | **W +** <br> M - | / |
| Sedentary | M + | / | / | / | / | / | / | / | *A +* | / | **W +** <br> **M -** | / |
| Overall | / | **W +** <br> **M +** <br> **A +** | / | / | *M + -* | / | / | / | / | / | **W +** <br> **M -** | / |
| *Lost job* | | | | | | | | | | | | |
| Active | / | / | / | / | / | / | / | / | / | / | / | / |
| Sedentary | / | / | / | / | / | / | / | / | / | / | / | / |
| Overall | / | **W +** <br> **M +** <br> **A +** | / | / | *M + -* | / | / | / | / | / | / | / |

"W" or "M" indicate result is applicable to women or men, respectively; "A" indicates results that are not stratified for sex/gender.

"Active" or "Sedentary" refer to occupation types related to specific estimates. Definitions of occupation types differed between studies (see S2 Table). "Active" refer to occupations held before the employment transition that may include physically demanding tasks or primarily standing work or manual work, while "Sedentary" refer to occupations that involve primarily sitting/desk work or no manual work. "Overall" indicates results are not stratified by occupation type. Results are based on fully-adjusted models from included studies (see S4 Table for covariates each study used).

**For weight changes relative to not retired/continued to work:** "+" indicates *higher or greater* body weight outcome relative to a referent group of continually employed or non-retired participants; "-" indicates *lower or lesser* body weight outcome relative to a referent group of continually employed or non-retired participants; "/" indicates no applicable results. Bolded symbols indicate a statistically significant comparison where *p*-values are at least 0.05 or smaller.

**For weight changes within employment transition (pre-post change):** "+" indicates *higher or greater* body weight outcome relative to baseline; "-" indicates *lower or lesser* body weight outcome relative to baseline; "/" indicates no applicable results. Bolded symbols indicate a statistically significant comparison where *p*-values are at least 0.05 or smaller. Italicized symbols indicate a reported pre-post result where statistical significance was not tested.

## Body weight outcomes within employment transitions by gender

A third (n = 4) of the included studies reported within-group weight change among the employment transitions [12,18,53,55]. Two of these studies reported results by occupation type only, which will be discussed in a subsequent section [18,55].

Among studies that reported the weight effect of retirement regardless of occupation type, women tended to experience weight gain while men both gained and lost weight. Monsivais

et al [12] found women gained an average of 0.48 kg per year (95% confidence interval (95% CI) 0.28 to 0.68) over a mean follow-up period of 3.5 years, and Stenholm et al [53] found women experienced a mean BMI increase of 0.15 kg/m$^2$ (95% CI 0.10 to 0.20) and had higher risk of obesity (RR = 1.15, 95% CI 1.09 to 1.21) in the survey wave right after retirement compared to the wave right before retirement (an average of 4 years apart between waves). However, among men, Monsivais et al [12] found men increased their weight when they entered retirement (0.52 kg/year, 95% CI 0.34 to 0.70) but Stenholm et al [53] found men experienced a decrease in BMI (-0.11 kg/m$^2$, 95% CI -0.22 to -0.01), although, the quality of the second study is lower than the first.

In addition to analyzing weight change following retirement, Monsivais et al [12] found women and men who lost their jobs also gained weight at an average of 0.69 kg (95% CI 0.46 to 0.92) and 0.68 kg (95% CI 0.43 to 0.92) per year, respectively.

In addition to the four studies that examined weight change following employment transitions, Morris et al [16] reported proportions of men who experienced weight gains or losses of more than 10% following job-loss and retirement; however, statistical comparisons were limited to across-group differences only. While the proportions are not discussed in this section, they can be found in S3 Table.

## Body weight outcomes of entering retirement compared to continued employment or non-retired groups by gender

Among five studies that reported results for women who retired relative to women who continued employment or were not otherwise retired [12,17,51,54,56], results were mixed: two studies found that retirement was associated with higher body weight outcomes [17,51], while three studies did not find that outcomes differed between those who retired and those who remained employed [12,54,56]. Using the U.S. Health and Retirement Study (HRS), Forman-Hoffman et al [17] found that among women with normal weight at baseline, retirees were more likely to gain 5% or more in BMI compared to those who continued to work at least 20 hours a week two years later (OR = 1.30, 95% CI 1.01 to 1.69). Pedron et al [51] used data from the KORA study in Germany and found that women gained 0.82kg/m$^2$ more compared to women who were not retired over a follow-up period of between 6 to 10 years.

Studies that did not find a statistically significant effect of retirement on weight outcomes had periods of follow-up ranging from 2 to 4 years [12,54,56], and settings generally differed from the two studies above, with one exception [56]. Godard [56] examined weight outcomes between retirees and those who continued to work using data from the Survey of Health, Ageing and Retirement in Europe (SHARE), which included Germany among seven other European countries. In contrast to the findings of Pedron et al [51], Godard [56] found a non-statistically significant increase of 0.31 kg/m$^2$ over a 2 to 4 year follow-up period. While Godard [56] used respondents who reported being employed or self-employed as their comparator, Pedron et al [51] included a broader group of non-retirees in more countries which included those who were unemployed and those experiencing long-term illnesses. However, the findings in Pedron et al [51] remained robust in sensitivity analyses when the comparator was limited to those who were employed. Of the three studies that did not find a statistically significant effect, two of the studies used objectively measured anthropometry [12,54], and one used self-reported body weight [56], while one of the two studies that found a statistically significant effect used self-reported body weight [17] and the other used objectively measured weight [51].

Results for men were similarly mixed. Among six studies that reported overall retirement effects for men [12,16,17,51,54,56], three studies found increased body weight outcomes

[16,54,56], and three did not find a statistically significant difference [12,17,51]. Of the three studies that found a statistically significant effect, one was rated as weak in quality (Feng et al [54]), one was rated moderate (Morris et al [16]) and one was rated as strong (Godard [56]), while the remainder of the three studies were all rated as moderate. In a study using data from the China Health and Retirement Longitudinal Study (CHARLS), Feng et al [54] found male retirees experienced weight gain of 2.10 kg (p<0.10) and a BMI increase of 0.92 kg/m$^2$ (p<0.05) higher compared to non-retirees over a two-year period based on objectively measured anthropometry. Morris et al [16] found that 7.5% of men who were retired for reasons unrelated to illness gained more than 10% of body weight five years later compared to 5.0% of men who were continuously employed (significantly different at p<0.05). While Godard [56] did not find men who retired experienced a significantly higher increase in BMI compared to those continuously employed, retirees did have a 11.5% increase in the probability of obesity compared to their working counterparts (p<0.05).

Comparing studies that reported a significant difference between men who retired relative to men who continued working or were otherwise unemployed with those that reported a non-significant difference in the same settings, some differences in analytic approach are notable. Monsivais et al [12] found men who entered retirement in the EPIC-Norfolk sample in the UK gained slightly less weight per year compared to those who remained employed, though the difference was not statistically significant. Model adjustments for important health behaviours relevant to weight outcomes, such as change in smoking and energy intake, were conducted in Monsivais et al [12]. This is in contrast to Morris et al [16], where adjustments for similar health behaviours were not done, and a significant difference was found. Morris et al [16] also found significantly more heavy drinkers and smokers among men who were not continuously employed compared to those who were, indicating non-negligible differences in the baseline characteristics of the two groups that may have explained the significant finding for body weight. Similar to the results among women discussed earlier, the results for men between Pedron et al [51] and Godard [56] were also mixed, but to a lesser extent. While Godard [56] found an increase in the probability of obesity among men, both studies reported non-significant results when the outcome variable was defined as BMI change [51,56]. Two of the three studies using objectively measured anthropometry did not report a significant finding [12,51], while two of the three studies using self-report did find that retirees differed in their weight gain from continued workers [16,56]. Where findings were statistically significant, the direction of change was generally consistent between studies using objective measurement and self-report [16,54,56].

## Body weight outcomes of job-loss compared to continued employment by gender

Among included studies, only two investigated transitions other than retirement [12,16], with one study sampling only men [16]. That study found that British men who became unemployed for reasons not due to illness or retirement were 7.1% more likely to gain over 10% in body weight 5 years later at follow-up compared to 5.0% among men who were continuously employed over 5 years (statistically different at p<0.05). The second study considered the effects of employment transitions on annualized weight gain separately for British women and men [12]. Monsivais et al [12] found that annualized weight gain was significantly higher among women who lost their job (0.72 kg/year) compared to those who remained employed (0.42 kg/year) (p = 0.007). For men, weight gain did not significantly differ between those who remained employed and those who lost their job (0.63 kg/year and 0.68 kg/year, respectively) [12].

## Body weight outcomes of employment transitions by occupation type and by gender

Seven studies stratified analyses by occupation types [14,17,18,52,53,55,56]. All studies defined occupation types based on the physical demands of the work, which often led to categorizing jobs as active versus sedentary, or manual versus professional/managerial (see S2 Table for our classification of results by occupation type).

Gueorguieva et al [55] did not stratify their results by sex/gender, but they found retirement was associated with yearly BMI increases that differ by occupation type (see S3 Table). Retirees from service and other blue-collar occupations tend to have a higher BMI per year change (0.12 kg/m$^2$, p<0.05, and 0.13 kg/m$^2$ per year, p<0.01, respectively) relative to retirees from professional and managerial occupations (0.04 kg/m$^2$ per year). Similarly, Chung et al [52] found weight gain among retires to be higher compared to those who were currently working in those occupations and subgroup analyses found this was primarily due to those retiring from physically demanding occupations (p<0.05).

Stratification of results by job-type accounted for some of the differences in the general effect of employment transitions on body weight by gender; however, findings remain mixed.

Weight gain in women who retired from a physically demanding job was reported in two studies [17,53]. Physically heavy work was associated with an increase in BMI of 0.30 kg/m$^2$ (95% CI 0.15 to 0.46 kg/m$^2$) and an increased risk of obesity (RR = 1.20, 95% CI 1.07 to 1.34) during retirement transition in a Finnish cohort [53]. In a US prospective cohort, female retirees from blue collar occupations were 1.58 times more likely to gain at least 5% in BMI relative to those who remained working in those occupations (OR = 1.58, 95% CI 1.13 to 2.21) [17]. Two additional studies found BMI increased following retirement from physically demanding jobs; however, findings were not statistically significant [14,56].

Among men, weight increases following retirement from a physically demanding job are supported by three studies (Zheng [14], Nooyens et al [18] and Godard [56]) and not supported by two others (Forman-Hoffman et al [17] and Stenholm et al [53]). Zheng [14] and Forman-Hoffman et al [17] both utilized a similar dataset derived from the HRS in the US but had different findings. This difference may be reconciled through a difference in the studied outcome. Forman-Hoffman et al [17] set a clinically significant threshold of a 5% increase in BMI over a 2 year period as their outcome, while Zheng [14] used change in BMI as a continuous variable in their analysis. In addition to finding retirees from physically demanding jobs gained more weight compared to those who were not retired, Zheng [14] also found that the BMI change experienced by retirees from strenuous jobs was significantly different from the BMI change experienced by retirees from sedentary jobs. Nooyens et al [18] examined change in weight (kg) and waist circumference (cm) per year and Stenholm et al [53] examined BMI change and relative risk of obesity. Nooyens et al [18] found that men who retired from active occupations experienced greater annualized weight gain and waist circumference growth compared to retirees from sedentary jobs. In contrast, Stenholm et al [53] found that men who retired from sedentary jobs experienced decreases in their BMI and retirement from physically demanding occupations did not significantly change BMI or risk of obesity. Finally, Godard [56] found that men retiring from strenuous occupations had an increased probability of obesity compared to those who were continuously employed, and that this increase was statistically significantly more than retirees from sedentary jobs.

## Role of sleep and other health behaviours as potential mediators

Only one study included analyses of sleep. Monsivais et al [12] reported those who experienced job-loss had greater odds of subsequent sleep loss due to worry relative to those who remained

employed (OR = 4.5, p<0.01). In contrast, the odds of sleep loss were not statistically different between those who entered retirement and those who remained employed [12]. Despite a lack of a formal analysis of mediation, Monsivais et al [12] suggests that sleep may play a mediating role in the relationship between employment transitions and body weight change for future research to consider.

In light of the limited evidence related to sleep, we further explored the included studies for other health behaviours that may mediate the impact of employment transitions on weight change. While most studies adjusted for health behaviours associated with body weight, several studies examined change in smoking and alcohol consumption, physical activity, and diet as outcomes following employment transitions [12,14,16–18,54,56], but only Nooyens et al [18] included separate models that examined the effect of change in health behaviours on weight change. None of the other included studies statistically examined health behaviours as mediators, but instead offered conceptual interpretations that the examined health behaviours may be potential mechanisms or channels through which employment transitions may impact subsequent weight change.

Morris et al [16] assessed changes in alcohol consumption and smoking following employment transitions, and found that men who became unemployed for reasons unrelated to illness were more likely to reduce their drinking compared to those who were continuously employed, but were not any more or less likely to reduce their cigarette smoking than continuously employed men. In the discussion of the results, Morris et al [16] did not link these findings with their weight change findings, but instead suggested that the weight gain observed among men who experienced job-loss may be due to changes in diet or physical activity.

Physical activity was the most common health behaviour examined [14,16–18,54,56]. An increase in physical activity post-retirement was commonly found; however, the findings by gender are mixed [14,17,18,54,56]. Godard [56] found women were more likely to engage in moderate *leisure-time* physical activity following retirement while Forman-Hoffman [17] and Zheng [14] both found that retirees did not engage in more vigorous physical activity. However, Forman-Hoffman [17] and Zheng [14] assessed change in vigorous physical activity as inclusive of work-based physical activity, which may mask how *leisure-time* physical activity changes following retirement. Among men, studies found an increase in vigorous physical activity [14,17]. Zheng [14] found that the increase in vigorous physical activity was mainly among those who retired from sedentary occupations. However, Nooyens et al [18] and Godard [56] did not find that men increased their *leisure-time* physical activity after retirement.

Diet was also assessed as a hypothetical mediator. Feng et al [54] found that men increased food consumption following retirement and suggested that this may explain why men experienced increases in BMI following retirement while women did not. Nooyens et al [18] found that male retirees increased their frequency of intake of vegetables and rice and decreased their intake of potatoes specifically. When stratified by occupation type, retirees from sedentary jobs also decreased their intake of proteins while retirees from active jobs decreased their overall energy intake and fibre density [18]. Of these changes in diet following retirement, only a decrease in fibre density was associated with subsequent weight gain [18].

## Discussion

In this systematic review, we screened 6,001 records and identified 12 studies that investigated the longitudinal association between employment transitions and body weight change in middle-aged and older adults. Few studies added to our understanding of job-loss and its effect on body weight as the most commonly examined transition was retirement. Many of the studies

reporting the effects of employment transitions on subsequent body weight change recognized that the physical demands of different occupations can influence body weight [14,17,18,52,53,55–57]. Almost all studies were in moderate quality and accounted for both gender and occupation type differences through study design or analyses. The results of the included studies generally fall into two categories: pre-post changes in weight within employment transitions or changes in weight for employment transitions relative to a comparison group (i.e., those who remained employed or did not retire). Findings from pre-post analyses suggest women tend to experience weight gain [12,53], and men may experience either weight loss [16,53], or weight gain following retirement [12,16,18], but there is much uncertainty associated with these findings due to the few pre-post only studies available and even fewer studies that included women in the study sample. When stratified by occupation type, weight gain may be related to retirement from physically demanding occupations [18,53], while weight loss may be related to retirement from sedentary occupations [53]. The modifying effect of occupation type, however, on weight change due to retirement is also likely to further differ by gender. As for example, men retiring from physically demanding occupations may lose, not gain, weight if that weight is predominantly muscle from occupations involving weight-bearing activity [58]. While these studies provide us with some insight into potential trajectories in weight change within employment transitions, evidence is more robust when a comparison group is used to determine weight change from employment transitions.

Among between-group studies, we found that women who enter retirement tended to experience more weight gain [17,51], or similar weight change, relative to their working or non-retired counterparts [12,54,56]. However, weight gain was not consistently supported among studies using objectively measured anthropometry [12,54]. The literature also appears inconsistent in reported effects of retirement for among men [12,17,51,54,56]. The evidence on gender differences, however, is not clear given that both the increases and decreases in the weight of women and men were not consistently found across different studies, potentially owing to differences in how studies defined their outcomes [12,14,17,51,54,56]. Moreover, we also found that occupation type modified the effect of retirement on weight, but results were more clear for the weight gain after retirement from physically demanding occupations [14,17,52,56], than the reported weight change following retirement from sedentary jobs [14,17,52,56]. Finally, job-loss appears to lead to more weight gain than continuous employment [12,16]; however, more studies are required to ascertain gender differences.

The findings of weight change among included studies may be relatively modest when compared to age-related weight trajectories of middle-aged and older adults [59–61]. Within the HRS, women and men generally experienced linear increases in BMI from ages 51 to 75 years [60]. Zheng [14] simulated BMI change 10 years post-retirement using the results from their analysis of retirees within the HRS for sedentary and strenuous occupations and found the weight loss experienced among retirees from sedentary occupations was reversed by age effects, leading to modest BMI gains post-retirement transition. However, this linear rise in BMI may not be true for settings outside of the USA. Using data from the Canadian National Population of Health Survey (NPHS), Wang et al [61] found women and men in Canada generally experience steady decreases in BMI from ages 65 to 96 years. Further, Dugravot et al [59] found women and men in the French GAZEL study experienced a gradually slower rate of BMI increase between the ages of 45 to 65 years, suggesting a plateauing of weight change. It is important to note the differences in how variables were defined across the included studies and how it impacts both the statistical and clinical significance of findings. For example, the inclusion of those who are long-term unemployed in a "not retired" group may not make it a suitable comparator as long-term unemployment may be associated with chronic illnesses that have consequences on body weight; however, Pedron et al [51] showed that their results were

statistically robust even with different definitions of their comparator. Operationalization of weight gain as a categorical or as a continuous variable, however, does seem to influence whether a statistically significant finding is detected, as was the case with Zheng [14] and Forman-Hoffman et al [17]. Change in BMI was measured on a continuous scale in Zheng [14] while change was categorized as 5% or greater gain, 5% or greater loss, or no change in Forman-Hoffman et al [17]. A statistically significant increase in BMI was found among retired men in Zheng [14], but the odds ratio for 5% or greater gain was not significant in Forman-Hoffman et al [17], suggesting men may gain weight following retirement that does not exceed clinical thresholds. Indeed, many of the findings of weight gain measured on a continuous scale are below clinically significant thresholds. Gains in excess of 1 kg/year have been associated with elevated risk of all-cause and cardiovascular disease-related mortality in older adults (age 50+) [62]. Similarly, changes of 5% in weight or +/- 1 BMI unit have also been found to be associated with excess mortality risk [63].

The difference in continuous and categorical measures of body weight in middle-aged and older adults is discussed in Paige et al [64]. In examining the effect of education level on annual weight change in a population-based cohort of adults aged 45 and older in Australia, Paige et al [64] defined weight change in four different ways: a categorical variable using percentage change, a categorical variable using absolute change, a continuous variable using percentage change, and a continuous variable using absolute change. The authors found no statistically significant or clinically meaningful change in body weight when the continuous variables were modeled [64]; however, they found that those with higher educational attainment had lower risk of both weight gain and loss of more than 1kg compared to those with no school certifications [64]. Future research that examines employment transition and weight change should consider a similar approach to modeling by including alternative measures and definitions of the outcome to avoid obscuring potentially important directions and clinically meaningful changes. Additionally, future literature reviews of this relationship should consider alternative methods to synthesize information from studies, such as individual participant data meta-analyses which can help address differences between studies in outcome definition, factors adjusted for, comparator groups used, and setting [65].

Our search only identified one study, that examined sleep in addition to the effect of employment transitions on body weight. While no other study included sleep in their analyses, there was some acknowledgement that sleep might also be an important underlying mechanism that has not yet been examined [51,56]. Both Godard [56] and Pedron et al [51] cited previous work from Eibich [66] who examined the effect of retirement on health and health behaviours. Eibich [66] found that improved overall health status following retirement was mediated through improvements in sleep and increases in physical activity. Monsivais et al [12] found that those who lost their job were more likely to experience sleep loss compared to those continuously employed. Taken together, sleep may be an important mediator for health outcomes across both involuntary transitions like job-loss and voluntary transitions like planned retirement. Given the paucity of studies that considered sleep and that the existing literature remains mixed with respect to mediation by physical activity and moderation by occupation type, future research ought to explore sleep further as suggested by authors in this area.

There are several limitations with this review. The first limitation is the difficulty in identifying studies that investigated "employment transitions" as this did not appear to be a term that was used explicitly in many articles. Often, a thorough reading of the methods section was required to verify whether a study truly examined a change in employment status over, at least, two points of time. However, our use of a dual independent reviewer approach and practice screenings would have mitigated most of this problem. However, some studies may still have been missed. A second limitation is related to our quality appraisal tool. Observational studies

that used either pre-post or pre-post with comparison were both given the same rating under the design domain, despite the latter offering potentially more robust findings when well-executed. Assigning a different score to each of the studies would have helped to further differentiate the quality of the evidence. Thirdly, publication bias may be present. As this review has found, null results have been observed when considerations around sex/gender, occupation type, and direction of body weight change are not adequately addressed. Lastly, the studies identified were mostly based in high-income countries. With respect to differences in labour force markets, cultural norms of working and ageing, health and pension systems, we expected the impacts of employment transitions on body weight to vary by countries. This highlights another important gap in the literature that future research should address.

The findings from our systematic review suggest that the effect of employment transitions on body weight outcomes in middle-aged and older adults differ by gender, occupation types and countries. The existing body of literature brings to attention the large research gaps. Potential key areas of focus in future studies include: the causal relationship and sleep as a third variable, the effect of job-loss as opposed to retirement, whether the effect of employment transitions differ by levels of socioeconomic status or across comorbidities related to overweight or obesity and, the impact of alternative operational definitions of weight change on study results. More rigorous evidence synthesis methods are also needed in future literature reviews to address study and topic complexity.

## Supporting information

**S1 Checklist.**
(DOCX)

**S1 Table. Search strings used in each bibliographic database.**
(DOCX)

**S2 Table. Details of employment categories in included studies.**
(DOCX)

**S3 Table. Results of included studies by sex/gender.**
(DOCX)

**S4 Table. Covariates used for main model in included studies.**
(DOCX)

## Acknowledgments

We would like to thank Ursula Ellis who supported the literature search process by providing invaluable feedback on our search strategy. Alexander C.T. Tam would like to acknowledge the support provided by the Canadian Institutes of Health Research (CIHR) through the Canada Graduate Scholarship–Master's, which supports Alexander's academic training. Rachel Murphy, Wei Zhang and Annalijn I. Conklin would like to acknowledge the Michael Smith Foundation for Health Research (MSFHR) for the support provided for their respective research programs through the MSFHR Scholar Award.

## Author Contributions

**Conceptualization:** Alexander C. T. Tam, Rachel A. Murphy, Wei Zhang, Annalijn I. Conklin.

**Data curation:** Alexander C. T. Tam, Veronica A. Steck, Sahib Janjua, Ting Yu Liu.

**Formal analysis:** Alexander C. T. Tam, Veronica A. Steck, Sahib Janjua, Ting Yu Liu.

**Investigation:** Alexander C. T. Tam, Veronica A. Steck, Sahib Janjua, Ting Yu Liu.

**Methodology:** Alexander C. T. Tam, Rachel A. Murphy, Wei Zhang, Annalijn I. Conklin.

**Project administration:** Alexander C. T. Tam, Annalijn I. Conklin.

**Resources:** Alexander C. T. Tam.

**Supervision:** Alexander C. T. Tam, Rachel A. Murphy, Wei Zhang, Annalijn I. Conklin.

**Validation:** Alexander C. T. Tam, Wei Zhang.

**Visualization:** Alexander C. T. Tam, Veronica A. Steck, Sahib Janjua, Ting Yu Liu, Annalijn I. Conklin.

**Writing – original draft:** Alexander C. T. Tam, Veronica A. Steck, Sahib Janjua, Ting Yu Liu.

**Writing – review & editing:** Rachel A. Murphy, Wei Zhang, Annalijn I. Conklin.

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
