## [Decision Letter · Decision Letter 0]

12 Apr 2022

PONE-D-21-39118A systematic review of evidence on employment transitions and weight change by gender in ageing populationsPLOS ONE

Dear Dr. Tam,

Thank you for submitting your manuscript to PLOS ONE. After careful consideration, we feel that it has merit but does not fully meet PLOS ONE’s publication criteria as it currently stands. Therefore, we invite you to submit a revised version of the manuscript that addresses the points raised during the review process.

Your paper has been thoroughly reviewed by two experts in the field and by myself. Their feedback is included in this email. At this time it is largely agreed that the paper provides interesting information regarding life transitions and changes in body mass and body mass index. The writing is generally clear and raises some important points regarding effect modifiers, gender, job type and outcome measurement. The reviewers have raised some important points that should be considered along with my feedback in your revisions. My specific feedback is listed below.

We look forward to receiving your revised manuscript.

Kind regards,

Stephanie Prince Ware, PhD

Academic Editor

PLOS ONE

Journal Requirements:

Additional Editor Comments (if provided):

The manuscript refers to body “weight” throughout, but the correct term is body mass. Please refer to this paper:

Winter EM, Abt G, Brookes FB, Challis JH, Fowler NE, Knudson DV, Knuttgen HG, Kraemer WJ, Lane AM, van Mechelen W, Morton RH, Newton RU, Williams C, Yeadon MR. Misuse of "Power" and Other Mechanical Terms in Sport and Exercise Science Research. J Strength Cond Res. 2016 Jan;30(1):292-300. doi: 10.1519/JSC.0000000000001101. PMID: 26529527.

Ensure that middle-aged is consistently hyphenated.Page 5, please include in brackets examples of the “other modifiable health behaviours”.Methods: Please provide the full bibliographic database-specific search strategies; these can be placed in a supplement.Was full-text screening also done by two independent reviewers?Can the authors please expand on what they mean by 11 of the 21 items were directly relevant to assigning a global rating for risk of bias? What did they do with the other 10 items?Was this review protocol registered or published prior to commencement? Please include the proposal registration information.Was a meta-analysis considered?Page 7, BMI is incorrectly spelt out as “body weight index” rather than “body mass index”Why were other measures of body mass status not incorporated (e.g., waist circumference, waist-to-hip ratio, body fat %)Table 4 – why was a p<0.10 chosen for identifying “significance”? Also, the text for Results includes other p-values (e.g., p<0.05, p<0.01).Table 4 – suggest changing O to A (all)Table 4 – Feng et al., why is there a + and – for women? Same for Forman-Hoffman et al. for men.It’s not clear if the findings are based on the statistical significance of null or fully adjusted models? Ideally they should be from fully adjusted models and the covariates included in the adjustment should be provided. Were comorbidities and socioeconomic status considered as potential effect modifiers?Page 19, the discussion of results for Monsivais et al. was a bit repetitive for men and women.Page 19, four studies regarding weight change following employment transitions are mentioned, but only one (Morris et al.) is discussed.Page 22, The first paragraph contains content that is best suited for the Introduction or Discussion.Page 24, this is a post-hoc analysis regarding other health behaviours. This should be outlined in the Methodology.The authors should interpret all results relative to the certainty of the evidence. Additionally, many of the broad stroke findings are related to 1 or 2 studies.The PRISMA checklist says the “certainty of evidence” was assessed, but there is no description of this in the Methods or the Results.The PRISMA checklist item 13f is missing a description.If the study protocol was not registered, this should be stated as a major limitation of the study.Was there any difference in outcomes based on whether body mass was self-reported or objectively measured?  There is no discussion about how this may affect results.

Reviewers' comments:

Reviewer's Responses to Questions

**Comments to the Author**

1. Is the manuscript technically sound, and do the data support the conclusions?

Reviewer #1: Yes

Reviewer #2: Yes

2. Has the statistical analysis been performed appropriately and rigorously? 

Reviewer #1: N/A

Reviewer #2: N/A

3. Have the authors made all data underlying the findings in their manuscript fully available?

Reviewer #1: Yes

Reviewer #2: Yes

4. Is the manuscript presented in an intelligible fashion and written in standard English?

Reviewer #1: Yes

Reviewer #2: Yes

5. Review Comments to the Author

Reviewer #1: This is a helpful review summarising the literature on this important topic. I have the following suggestion to improve the manuscript.

Abstract:

'Employment is strongly associated with weight' is a very general statement, and does not tell us anything about the direction of the association. Maybe this could be more specific?

'they are also more vulnerable to employment transitions' Since employment transitions have not be defined, it may be more helpful to say 'job loss'.

Introduction:

line 9 'lower incidence' may not be the appropriate term for education and skills training.

It would be helpful early on to define the term 'employment transitions' and introduce the range of employment transitions that are of interest.

Methods:

Was the review protocol registered before starting - if yes please add details in the methods section.

It may be helpful for clarity to add an inclusion/exclusion table that follows a standard format e.g. Setting/Participants/Exposure/Outcomes/Study type/Publication type/Publication year/Language.

Results:

Further details need to be added to Table 4 so that this table can be understood without reference to the article text. For example 'Active, Sedentary, Overall' presumably refer to the type of job that the participant held before retirement or job loss, but this is not stated. Also some details should be provided in the methods of how jobs were categorised into active or sedentary, - the type of information which was extracted from the papers and how this was split into the categories presented.

Table 4 could be a very helpful overview of the evidence, but needs a little work to make it more easy to understand.

Conclusions

Given the wide range of factors that may contribute to changes in weight status at job loss or retirement, it may be worth considering an individual participant data meta-analysis in future work. (ref: https://www.bmj.com/content/340/bmj.c221)

Reviewer #2: This is a relevant review investigating the impact of employment transitions on body weight. Twelve studies were included in the review, all considering transitions from employment to retirement, however only two studies considered transitions from employment to unemployment. The conclusion of the review is that the existing studies do not provide a clear enough picture to draw any conclusions on the impact of employment transitions and weight change.

Overall, the review is well written, and I believe the topic is relevant and the review will contribute to the literature by highlighting important areas where more research is needed.

Comment #1

The conclusion is that no firm conclusions can be drawn, however, when looking at table 4, it seems pretty consistent that women gain weight after retirement. Do you believe that there is not enough evidence to draw this conclusion?

Comment #2

Odds ratios are not presented in the same way throughout the manuscript. Please try to align this.

Comment #3

You assess the quality of each study; however, you do not mention this when describing the results or when you draw conclusions. It would be nice to see some considerations of the quality – do the good quality studies weigh more when drawing conclusions? Or do you weigh the results from each paper equally?

Comment #4

Introduction page 3: I do not understand the sentence: Coupled with the rise in labour force participation of middle-aged and older workers are unique challenges such as lower incidence of continued education and skills training. Perhaps you could elaborate why this is relevant.

Comment #5

Methods page 5: The sentence: Screening, assessment, and inclusion of studies published in English, French, and Chinese. I feel like there is something missing to this sentence?

Comment #6

Results page 7: you write body weight index, but it should be body mass index.

Same page: The sentence: odds or risk of becoming overweight or obese. There is a consensus within the research field of obesity that obesity is not something you are or become – it is a condition that you have. Like a disease. Whatever disease a person may have, it may not define them as persons or individuals. For example, having a chronic disease such as cancer does not make a person identify as a cancerous person. I would rewrite the sentence: odds or risk of overweight or obesity. Please go through the manuscript to make sure you use person-first language throughout.

Comment #7

Table 2: It should be outcome(s) measures instead of outcome(s) measured. Sometimes you use a full stop and sometimes you don´t.

Comment #8

A few times (e.g. on the bottom of page 19) you describe women who continued employment or were not otherwise retired/unemployed. Please elaborate what you mean by not otherwise retired/unemployed.

Comment #9

Results page 24: Suggestion for rephrasing the sentence: suggests that there may be a mediating role that sleep has in the impact of employment transitions on body weight change… to: suggests that sleep may play a mediating role in the relationship between employment transitions and body weight change…

Same page: you write: dietas outcomes. I have never come across this word before. Is it a typo or is it a real phrase?

Comment #10

Discussion page 26: you write: However, this linear rise in BMI may not be true for other countries. Perhaps you should mention here where the study is conducted, as this is not fresh in the readers memory.

Comment #11

Discussion page 27: you mention that operationalization of weight gain as a categorial or continuous variable has influenced the results in the studies by Zheng and by Forman-Hoffman et al. It seems that you only describe changes in BMI as a continuous variable and do not mention the categorical changes? I do not see from your discussion how categorical or continuous weight change has influenced the results.

Additionally, in the following section, you mention a study examining weight change as both categorical and continuous (Paige et al.); however, you only mention the results for the continuous variable.

I am a bit confused what you wish to discuss with these two paragraphs, and I do not think that the message is coming across very clear. Perhaps you should consider rewriting these paragraphs.

6. PLOS authors have the option to publish the peer review history of their article (what does this mean?). If published, this will include your full peer review and any attached files.

Reviewer #1: No

Reviewer #2: No

---

## [Author Response · Author response to Decision Letter 0]

27 May 2022

We have copied our responses from the attached Response to Reviewer document into this field. We wish to thank the editor and the reviewers again for their careful reading of our manuscript.

*Please note the page numbers referenced are mapped to the clean version of the manuscript.

Journal Requirements:

 and 

We have revised our manuscript in accordance with the style requirements and file naming conventions.

*** Editor comments ***

1. The manuscript refers to body “weight” throughout, but the correct term is body mass. Please refer to this paper:

Winter EM, Abt G, Brookes FB, Challis JH, Fowler NE, Knudson DV, Knuttgen HG, Kraemer WJ, Lane AM, van Mechelen W, Morton RH, Newton RU, Williams C, Yeadon MR. Misuse of "Power" and Other Mechanical Terms in Sport and Exercise Science Research. J Strength Cond Res. 2016 Jan;30(1):292-300. doi: 10.1519/JSC.0000000000001101. PMID: 26529527.

Thank you for sharing an article that highlights the difference between body mass and Newtonian weight and the continued incorrect usage of “weight” in sport and exercise science. We acknowledge that the use of body weight in our manuscript to refer to measures of kilograms or body matter would be imprecise in the Winter et al. sense. In our experience, within obesity epidemiology, the discourse around changes to anthropometrics (measured in e.g., kg/BMI) primarily expresses it as changes to body weight, and not as body mass – see examples from Hu et al 2018; Mozaffarian et al 2011; and Tasali et al 2022.

The use of body weight as the term used to capture kilograms also bears clinical relevance. For example, clinical practice guidelines for managing overweight and obesity from North American and European clinical professional groups make similar statements using body weight rather than body mass when referring to anthropometric changes. Guidelines have also identified clinically meaningful thresholds for weight change targets to manage cardiovascular and metabolic risks and other comorbidities; gaining 5 kg or more in body weight during adulthood is a risk for many major chronic conditions (WHO 2004). 

We acknowledge that PLOS One appeals to a wide variety of disciplines and to both clinical and non-clinical audiences alike. We believe that the term “weight” is the more accessible word choice; however, we leave it the editor to make the final decision regarding the use of “weight” or “mass” in our manuscript. 

American Psychological Association, Clinical Practice Guideline Panel. (2018). Clinical practice guideline for multicomponent behavioral treatment of obesity and overweight in children and adolescents: Current state of the evidence and research needs. Retrieved from http://www.apa.org/obesity-guideline/obesity.pdf

Hu Y, Zong G, Liu G, et al. Smoking Cessation, Weight Change, Type 2 Diabetes, and Mortality. N Engl J Med. 2018;379(7):623-632. doi:10.1056/NEJMoa1803626

Mozaffarian D, Hao T, Rimm EB, Willett WC, Hu FB. Changes in diet and lifestyle and long-term weight gain in women and men. N Engl J Med. 2011;364(25):2392-2404. doi:10.1056/NEJMoa1014296

Tasali E, Wroblewski K, Kahn E, Kilkus J, Schoeller DA. Effect of Sleep Extension on Objectively Assessed Energy Intake Among Adults With Overweight in Real-life Settings: A Randomized Clinical Trial. JAMA Intern Med. 2022;182(4):365-374. doi:10.1001/jamainternmed.2021.8098

Wharton S, Lau DCW, Vallis M, et al. Obesity in adults: a clinical practice guideline. CMAJ. 2020;192(31):E875-E891. doi:10.1503/cmaj.191707

Yumuk V, Tsigos C, Fried M, et al. European Guidelines for Obesity Management in Adults [published correction appears in Obes Facts. 2016;9(1):64]. Obes Facts. 2015;8(6):402-424. doi:10.1159/000442721

World Health Organization. Global strategy on diet, physical activity and health. Geneva, Switzerland: WHO Publications, 2004

2. Ensure that middle-aged is consistently hyphenated.

We have made edits to consistently hyphenate middle-aged throughout the manuscript.

3. Page 5, please include in brackets examples of the “other modifiable health behaviours”.

We have provided examples of the health behaviours that the cited studies used. The sentence now reads:

Sleep may contribute to explaining the heterogeneity that appears to exist in the literature on job-loss and retirement and body weight gain in older adults, yet it has not received as much attention as physical activity [14,18], or other modifiable health behaviours (e.g., alcohol consumption and smoking) [15,16].

4. Methods: Please provide the full bibliographic database-specific search strategies; these can be placed in a supplement.

We have placed the search strategies into a supplemental table (new Table S1). We have referenced it in our Methods section.

5. Was full-text screening also done by two independent reviewers?

Yes, full-text screening (as well as reference-tracing of included studies) was also done by two independent reviewers following the same division of work as with title and abstract screening. We have updated the manuscript to clarify this. The change can be found on page 8:

“Records were removed based on exclusion criteria and eligible full-texts were retrieved and screened for inclusion and reference-tracing adhering to the same division of work between two independent reviewers.”

6. Can the authors please expand on what they mean by 11 of the 21 items were directly relevant to assigning a global rating for risk of bias? What did they do with the other 10 items?

Thank you for the opportunity to clarify. Within the quality appraisal tool that we selected, there are 11 items that the authors of the tool deem to be relevant to their scoring algorithm. These 11 questions are presented in Table 3. In addition to these items, there are a total of 10 additional questions in the tool that have no bearing on how the studies are scored. The topics covered by these questions are described below. We excluded presenting the responses to these items as the core focus of the excluded items pertain to clinical trials, which our responses to would be “not applicable” and are not consequential to the final rating.

Section B asks 3 questions specific RCTs and whether the selected design was appropriate for the research question.

Section G asks 3 questions regarding the integrity of the intervention received (e.g., percentage of participants who were given treatment, likelihood of contamination).

Section H asks 4 questions regarding the appropriateness of the analytic approach selected by the authors (e.g., unit of analysis, intention to treat).

We have updated the manuscript with text to clarify why these 10 items were not relevant and not used. The text is on page 10 and reads:

“The remaining 10 items from the tool are not relevant to the studies that were included in our systematic review and thus, had no bearing on the scoring assignment. For example, the items include questions related to randomized controlled trial design, the integrity of the intervention received, and intention-to-treat analytic approaches [34].”

7. Was this review protocol registered or published prior to commencement? Please include the proposal registration information.

The proposal was not registered prior commencement. Literature review work had begun as a part of ACTT’s thesis as background and rationale for the research and the suggestion to complete the work as a systematic review was made after. ACTT wrote a PROSPERO-style protocol document which was reviewed by the supervisory committee (listed as co-authors) and used to guide the actual systematic review work; however, we did not register it to the PROSPERO database. 

8. Was a meta-analysis considered?

We did not consider a meta-analysis. Based on some limited literature searches conducted by ACTT for his thesis proposal, we did not believe that we would have identified enough studies to allow for a quantitative synthesis like a meta-analysis. This was confirmed after only 12 studies were identified in the systematic review. Also, the included studies varied in how outcomes were defined (e.g., weight change, BMI change, categorical weight change of 1kg or 5%) and which subgroups were analysed, if any (gender or type of occupation). Therefore, our strategy for synthesizing results was narrative in nature. Our findings showed wide heterogeneity and paucity of data that would confirm this choice.

9. Page 7, BMI is incorrectly spelt out as “body weight index” rather than “body mass index”

We have corrected this typographical error.

10. Why were other measures of body mass status not incorporated (e.g., waist circumference, waist-to-hip ratio, body fat %)

We were interested in change in body weight over time given the important role that excess weight gain of 5 kg or more has for chronic disease (WHO 2004). While we did not explicitly exclude other measures of body weight, our search terms were selected to reflect our research question: ‘what is the impact of employment transitions on body weight in women and men?’ Anthropometric measures such as waist circumference, waist-to-hip ratio and body fat % are other important obesity outcomes that reflect body composition and could be the focus of a future review. We note, however, that we did find papers that included such anthropometric outcomes in addition to Kg. 

World Health Organization. Global strategy on diet, physical activity and health. Geneva, Switzerland: WHO Publications, 2004

11. Table 4 – why was a p<0.10 chosen for identifying “significance”? Also, the text for Results includes other p-values (e.g., p<0.05, p<0.01).

The use of p<0.10 in Table 5 (formerly Table 4, please see response to reviewer #1’s comment on our Methods) was originally based on some of the included studies where the selected and 

presented p-value was p<0.10, so we applied this cut-off to all studies. However, after discussion and re-consideration, we have decided to update this table to reflect results that were significant at p<0.05. The table description has been updated to reflect the new cut-off, and the results have been cross-checked with the studies in line with the new p-value.

We wish to clarify that the p-values reported in the Results text refer to statistical significance level chosen by the authors of the included studies that met our review criteria. Thus, the range of values reflects the diversity in reporting of significant findings and thus the overall evidence. 

12. Table 4 – suggest changing O to A (all)

We have updated Table 5 (formerly Table 4) using the suggestion of A to represent “all” instead of O for “overall”. The note underneath Table 5 has also been updated to reflect the updated symbol.

13. Table 4 – Feng et al., why is there a + and – for women? Same for Forman-Hoffman et al. for men.

In Feng et al., the authors modelled both BMI and body weight in kilograms. For women, the BMI model showed weight gain of +0.25 kg/m2 while the kilogram model showed weight loss (-0.06 kg). Both results were not statistically significant at the authors’ chosen value of p<0.10. Therefore, we indicated this by using both the + and the – symbols for retired women, without boldface.

In Forman-Hoffman et al., the results are based on a multinomial logistic regression model using a three-level categorical outcome of ≥5% weight gain, ≥5% weight loss, and no change serving as the reference. The authors reported an odds ratio above 1 for both losing ≥5% and gaining ≥5% in weight among retiring normal weight men compared to men who continued to work, but the ORs were not significant at the authors’ chosen value of p<0.05. Therefore, we indicated this by using both the + and – symbols for men who retired, without boldface.

Detailed study results from each study can be found in Table S3 (formerly Table S2).

14. It’s not clear if the findings are based on the statistical significance of null or fully adjusted models? Ideally they should be from fully adjusted models and the covariates included in the adjustment should be provided. Were comorbidities and socioeconomic status considered as potential effect modifiers?

We agree and confirm that the findings reported are based on fully-adjusted models reported by the included studies. We have added a note to the bottom of Table 5 (formerly Table 4) and Table S3 (formerly Table S2) to indicate this. We have placed the covariates from the adjusted models into a supplementary table (Table S4).

Socioeconomic status variables were often considered by included studies only as confounders. We did not find the use of socioeconomic status variables as effect modifiers in any of the included studies. We had not previously explicitly investigated comorbidities as a potential effect modifier; however, we have revisited the included studies and we can confirm that the studies did not consider this either.

We would like to mention that Morris et al was perhaps the closest to considering effect modification by underlying disease. Their retired and unemployed groups were further stratified by whether the reason for the employment status change was due to illness or not; however, this was based on a self-reported reason for not working. The results for both reasons for employment transitions were placed in Table S3 along with the rest of the extracted results. We also added a comment in our Discussion as an area of future investigation. It can be found on page 37:

“Potential key areas of focus in future studies include: the causal relationship and sleep as a third variable, the effect of job-loss as opposed to retirement, whether the effect of employment transitions differ by levels of socioeconomic status or across comorbidities related to overweight or obesity and, the impact of alternative operational definitions of weight change on study results.”

15. Page 19, the discussion of results for Monsivais et al. was a bit repetitive for men and women.

Thank you for identifying an opportunity to improve the readability of the manuscript. We understand that this section uses repetitive language when presenting results. To rectify this, we have retained the language for the retirement results, but we have simplified the language in subsequent passages that cover results for job-loss. It can be found in the lower half of page 23: 

“In addition to analyzing weight change following retirement, Monsivais et al [12] found women and men who lost their jobs also gained weight at an average of 0.69 kg (95% CI 0.46 to 0.92) and 0.68 kg (95% CI 0.43 to 0.92) per year, respectively.”

16. Page 19, four studies regarding weight change following employment transitions are mentioned, but only one (Morris et al.) is discussed.

In the section that describes results “within employment transitions”, we limited our discussion to Monsivais et al and Stenholm et al because both studies present results for employment transitions without occupation types as effect modifiers. The two other studies that we counted among the four were Nooyens et al and Gueorguieva et al and they did not present results of their analysis for employment transitions overall, opting to only present results specific to certain occupation types. We decided to save the discussion of these two studies for a later section where we explore differences in outcomes by occupation type. We mentioned we would revisit Nooyens et al and Gueorguieva in the last sentence of the first paragraph on the top of page 23: “Two of these studies reported results by occupation type only, which will be discussed in a subsequent section […]”. The second paragraph does cover the results of Monsivais et al and Stenholm et al.

The Morris et al paper was not counted among the four papers of this section as their analysis compared across group differences (e.g., discontinuously employed vs continuously employed), rather than within each transition. However, the authors did present pre and post values where we could get a sense of the direction of change. We opted to discuss these briefly and reserved the majority of the discussion of across-group differences for a later section. We can remove it from the manuscript, if the editor considers this paragraph a distraction to the reader

17. Page 22, The first paragraph contains content that is best suited for the Introduction or Discussion.

We have deleted this paragraph and moved content of it into the first paragraph of the Discussion section as suggested.

18. Page 24, this is a post-hoc analysis regarding other health behaviours. This should be outlined in the Methodology.

We agree that the component of the synthesis related to other health behaviours beyond sleep is post-hoc, and we have updated the Methodology section with text that describes this additional work we carried out. It has been added to the bottom of page 10 and is provided here for reference:

“The narrative synthesis revealed additional health behaviours beyond sleep that were studied as mechanisms; thus, a post-hoc analysis was conducted to summarise additional results.”

19. The authors should interpret all results relative to the certainty of the evidence. Additionally, many of the broad stroke findings are related to 1 or 2 studies.

We agree that our broad stroke summary of findings for within-group pre-post changes in the Discussion section were originally related to only 1 or 2 studies. However, we believe that the paragraph that follows, regarding across-group differences, does sufficiently cover a wider range of studies (2 to 6 studies). We owe this to our search identifying more across-group studies (e.g., retired vs not retired) than pre-post/within-group studies, which enabled us to summarise findings from multiple studies that can be more general. More so, with respect to job-loss, we identified one study that included both women and men and a second study that only included men. Thus, we were limited again by the few studies that addressed this research area. To reflect the uncertainty due to limited evidence and the instance where we only connected our broad stroke findings to 1 or 2 studies, we have added new text on page 32 as follows:

“Findings from pre-post analyses suggest women tend to experience weight gain[12,53], and men may experience either weight loss [16,53], or weight gain following retirement [12,16,18], but there is much uncertainty associated with these findings due to the few pre-post only studies available and even fewer studies that included women in the study sample.”

We only included one sentence regarding the findings for job-loss, and we feel that it accurately highlights what we found in the two studies, so we have not edited it. We have provided an excerpt from page 33: “Finally, job-loss appears to lead to more weight gain than continuous employment [12,16]; however, more studies are required to ascertain gender differences.”

20. The PRISMA checklist says the “certainty of evidence” was assessed, but there is no description of this in the Methods or the Results.

Thank you for the opportunity to clarify. It was not possible to assess the certainty of evidence because the outcomes assessed – BMI, kg, kg/year, and odds of change – differed between studies. As noted earlier, we believed that generating a single estimate of effect would not be feasible and thus no measurement of certainty. 

We elected to, instead, use the Discussion section to re-present the conflicting findings and the methodological differences between studies that might explain mixed findings. This is what we wanted to indicate in the PRISMA checklist file by stating “N/A – narrative synthesis” and “Acknowledged mixed results.”. We have re-written the description in the PRISMA checklist (item 15 and 22) to the following:

“No certainty assessment was conducted for our narrative synthesis. We discuss the mixed results in the Discussion.”. For item 15, this is followed by some excerpts from the text that reflect this discussion.

21. The PRISMA checklist item 13f is missing a description.

We apologise for missing the description for item 13f in the PRISMA checklist. A description has been added. We have provided that text here for reference:

“No sensitivity analyses were conducted for our narrative synthesis.”

22. If the study protocol was not registered, this should be stated as a major limitation of the study.

We did not register the study protocol; however, we do not consider this to be a limitation in any way since registration in PROSPERO or OSF is not peer-reviewed. Furthermore, we followed a strict protocol for the search that was a priori and hence criteria for inclusion/exclusion and search terms are defined beforehand to ensure accurate, precise and repeatable searching and results of this review; we pilot-tested our search strategy with preliminary searches before finalising the protocol for the final search. A great deal of oversight was provided to this process by ACTT’s thesis supervisory committee (listed as co-authors: RAM, WZ, and AIC) of experienced researchers with topic expertise. In addition to ACTT’s checks, WZ also provided a secondary high-touch review of extracted information to ensure accuracy of the independent reviewers’ work. ACTT also consulted with a health science reference librarian at their institution in organizing this review. We would defer to the editor’s decision on how this presents a limitation. 

23. Was there any difference in outcomes based on whether body mass was self-reported or objectively measured? There is no discussion about how this may affect results

Thank you for the question. We had not considered this angle of the results, despite capturing which studies used self-report versus objectively measured outcomes in our characteristics table. To better highlight this, we have revisited our Table 5 (formerly Table 4) and added two headers to separate the studies that used measured outcomes and the studies that used self-report. The studies have been re-ordered under their appropriate header.

We have also updated our manuscript with some new text to explain some of the differences between studies using self-report vs objectively measured when comparing across employment transition groups. For men, 2/3 studies using self-report found a statistically significant effect of retirement on the outcome and 1/3 studies using objectively measured outcomes found a statistically significant effect. For women, there was also a difference between the two measurements: 1/3 studies using objective measurement found a statistically significant effect and 1/2 studies using self-report found a statistically significant effect. The new text can be found on pages 25 through 26 of the results and we added one comment to our discussion on page 33.

*** Reviewer #1 comments ***

Abstract:

'Employment is strongly associated with weight' is a very general statement, and does not tell us anything about the direction of the association. Maybe this could be more specific?

'they are also more vulnerable to employment transitions' Since employment transitions have not be defined, it may be more helpful to say 'job loss'.

We have revisited the background section of our abstract. The direction of association is bidirectional, and the direction of the change in weight is also dependent on the employment exposure itself. To give the readers something more specific, we have re-written the first sentence to reflect one area within the employment-weight relationship.

“Becoming unemployed is associated with poorer health, including weight gain.”

With respect to the use of “employment transitions”, we wished to capture job-loss, involuntary retirement, and other changes to employment status in a succinct term. However, as the reviewer has identified, we have yet to define the term within the abstract. We have replaced “employment transitions” with “changes to employment status” instead, which should help us retain our intention to refer to multiple kinds of transitions that can occur. The sentence now reads:

“Middle- and older-age adults are a growing segment of workforces globally, but they are also more vulnerable to changes to employment status, especially during economic shocks.”

Introduction:

line 9 'lower incidence' may not be the appropriate term for education and skills training.

It would be helpful early on to define the term 'employment transitions' and introduce the range of employment transitions that are of interest.

We have revisited the use of the term “lower incidence” from the cited work and have come up with “enrolment” as a more appropriate term to capture new entrants into skills training programs. Here is an excerpt of the updated text for your reference:

“Coupled with the rise in labour force participation of middle-aged and older workers are unique challenges such as lower rates of enrolment into continued education and skills training [5], employer preference for hiring younger employees [5], and ageism at work [5,6].”

We agree with the reviewer that we should introduce employment transitions earlier. We have added the following text when we first discuss the differential impacts of job-loss and retirement in our Introduction. Here is an excerpt:

“One of the risk factors for cardiovascular disease that has received some attention in the employment literature is body weight change [12–16], and how it differs following different changes in employment status. Changes in employment status, or employment transitions, span a range of events from exiting the labour force through job-loss and retirement to re-entry into the labour force by re-employment. Loss of a job without compensation has been found to be associated with […]”

Methods:

Was the review protocol registered before starting - if yes please add details in the methods section. It may be helpful for clarity to add an inclusion/exclusion table that follows a standard format e.g. Setting/Participants/Exposure/Outcomes/Study type/Publication type/Publication year/Language.

As indicated above, our review protocol was not registered before starting thus there is no number to add to the manuscript However, we did establish a clear protocol following standard PROSPERO procedures to ensure accurate, precise and repeatable searching and results of this review: 1) a priori criteria for inclusion/exclusion and search terms defined beforehand; 2) pilot-testing of search strategy with preliminary searches before finalising the protocol for the final search; and 3) separate searches and review by two independent researchers. Following this Reviewer’s suggestion, we have also summarised inclusion and exclusion criteria as the new Table 2. We have offset the subsequent table numbers (characteristics of included studies, quality appraisal, and summary of results) by 1 to accommodate this addition.

Results:

Further details need to be added to Table 4 so that this table can be understood without reference to the article text. For example 'Active, Sedentary, Overall' presumably refer to the type of job that the participant held before retirement or job loss, but this is not stated. Also some details should be provided in the methods of how jobs were categorised into active or sedentary, - the type of information which was extracted from the papers and how this was split into the categories presented.

Table 4 could be a very helpful overview of the evidence, but needs a little work to make it more easy to understand.

We thank the reviewer for sharing useful ways to improve the content of Table 5 (formerly Table 4). As suggested, we have added more details that allow Table 5 to stand alone without reference to manuscript text.

We have updated the title (new text in red) to better set up the reader for the content that it contains.

“Table 5. Summarised results from included studies by study design, employment transition, and occupation type.”

We have added a description of the occupation types presented in the table to the note under the table. Exact classifications of “active” or “sedentary” occupations on a per study basis are provided in full in a separate table (Table S2), but we have provided the breadth of criteria that were used to categorise them into the table as a note in Table 5.

‘“Active” or “Sedentary” refer to occupation types related to specific estimates. Definitions of occupation types differed between studies (see Table S2). “Active” refer to occupations held before the employment transition that may include physically demanding tasks or primarily standing work or manual work, while “Sedentary” refer to occupations that involve primarily sitting/desk work or no manual work. “Overall” indicates results are not stratified by occupation type.’

Conclusions

Given the wide range of factors that may contribute to changes in weight status at job loss or retirement, it may be worth considering an individual participant data meta-analysis in future work. (ref: https://www.bmj.com/content/340/bmj.c221)

We thank the reviewer for highlighting a methodological consideration for future studies. We agree that while resource intensive, the output of such a meta-analysis would be useful in processing the many factors that contribute to weight change following employment status change. We have added a line to our discussion to reflect this. Here is the text for your reference:

“Additionally, future literature reviews of this relationship should consider alternative methods to synthesize information from studies, such as individual participant data meta-analyses which can help address differences between studies in outcome definition, factors adjusted for, comparator groups used, and setting [66].”

*** Reviewer #2 comments ***

Comment #1

The conclusion is that no firm conclusions can be drawn, however, when looking at table 4, it seems pretty consistent that women gain weight after retirement. Do you believe that there is not enough evidence to draw this conclusion?

We agree that it may appear there is consistency results suggesting women gain weight after retirement. However, not all these studies can be directly compared, and we believe there is not enough evidence to draw a firm conclusion that women who enter retirement gain weight (pre-post) because only two studies provided estimates for women – one of which was rated as a weak study because of incomplete reporting of methods and sample selection. When we look at comparative studies (retirees vs continued workers), we also hesitate to draw any conclusions given the inconsistency in the weight gain finding. While the general direction is indeed positive, few studies found that the weight gain was statistically more than those who continued to work. For non statistically significant point estimates, they were generally small and close to the null; therefore, we are also unsure if drawing a conclusion on the direction (more vs less) is appropriate given current evidence. 

While we spend an appreciable amount of space in the Discussion section trying to unpack differences between studies that may help us contextualize why there were different findings, we feel as though our final consensus remains and we cannot say for certain that the retirement transition is associated with more gain than continuing to work for women. We think that a “no firm conclusion” conclusion accurately represents the truth of our review’s findings.

Comment #2

Odds ratios are not presented in the same way throughout the manuscript. Please try to align this.

We thank the reviewer for identifying misalignments in the presentation of odds ratios. We have gone through the manuscript to align them to the format of (OR = ##.##, 95% CI ##.## to ##.##) where possible. There is one instance where the study only provided a p-value for the OR, so we deviate from the format in that case and present the p-value in place of the confidence interval.

Comment #3

You assess the quality of each study; however, you do not mention this when describing the results or when you draw conclusions. It would be nice to see some considerations of the quality – do the good quality studies weigh more when drawing conclusions? Or do you weigh the results from each paper equally?

With respect to the conclusions drawn, we do not find that the quality changes the uncertainty that we have about the evidence. Findings from the weak studies are corroborated by findings from some of the moderate studies but not by others. For example, Feng et al (a study we rated as weak) found that men who retired experienced weight gain that was statistically significantly more than men who kept working. Godard (a study we rated as strong) and Morris et al (moderate) also found men gained weight, yet three moderate quality studies did not find this to be the case.

To bring some of this to the attention of the reader, we have added the above example into our manuscript. It can be found on page 25:

“Of the three studies that found a statistically significant effect, one was rated as weak in quality (Feng et al [54]), one was rated moderate (Morris et al [16]) and one was rated as strong (Godard [56]), while the remainder of the three studies were all rated as moderate.”

Comment #4

Introduction page 3: I do not understand the sentence: Coupled with the rise in labour force participation of middle-aged and older workers are unique challenges such as lower incidence of continued education and skills training. Perhaps you could elaborate why this is relevant.

We thank the reviewer for giving us an opportunity to clarify. We wanted to set the stage for why we should give attention to employment transitions in the age group (adults middle-aged and older) that this review targets. Despite the rise in labour participation of middle-aged and older workers, there are barriers for them to remain in the labour force. We do this by highlighting a few barriers that this age group encounters in the labour market.

The barrier that has been highlighted here is related to continuing education and skills development. Given the rapid advancement of technology, there is a need for workers to constantly maintain and even upgrade their skills over their working lives. Generally, middle-aged and older workers have lower participation rates in continued training compared to workers who are younger. There is also a disparity between employed and unemployed middle-aged and older workers where employed individuals have a higher participation rate – likely attributable to employer-sponsored training programs. This puts middle-aged and older workers at risk for job loss initially and at a slower rate of re-entry into the labour market compared to young age groups.

Matteo Picchio, 2015. "Is training effective for older workers?," IZA World of Labor, Institute of Labor Economics (IZA), pages 121-121, January.

Organisation for Economic Co-operation and Development. Employment Outlook 1998. Paris; 1998 Jun. Available: https://doi.org/10.1787/empl_outlook-1998-en

Comment #5

Methods page 5: The sentence: Screening, assessment, and inclusion of studies published in English, French, and Chinese. I feel like there is something missing to this sentence?

We have corrected this incomplete sentence. The sentence now reads:

“Screening, assessment, and inclusion were limited to studies published in English, French, and Chinese.”

Comment #6

Results page 7: you write body weight index, but it should be body mass index.

Same page: The sentence: odds or risk of becoming overweight or obese. There is a consensus within the research field of obesity that obesity is not something you are or become – it is a condition that you have. Like a disease. Whatever disease a person may have, it may not define them as persons or individuals. For example, having a chronic disease such as cancer does not make a person identify as a cancerous person. I would rewrite the sentence: odds or risk of overweight or obesity. Please go through the manuscript to make sure you use person-first language throughout.

We agree with the review to use first-person language and we have reviewed our manuscript to ensure this is the case throughout. Thank you for noting this.

The BMI typographical error has been corrected.

Comment #7

Table 2: It should be outcome(s) measures instead of outcome(s) measured. Sometimes you use a full stop and sometimes you don´t.

We changed this to Outcome measures since double plural is redundant.

We have removed the use of full stops and corrected inconsistent capitalization throughout Table 3 (formerly Table 2).

Comment #8

A few times (e.g. on the bottom of page 19) you describe women who continued employment or were not otherwise retired/unemployed. Please elaborate what you mean by not otherwise retired/unemployed.

We thank the reviewer for giving us an opportunity to clarify. The comparator group differed slightly between studies. While most studies only had continuously employed individuals serving as their comparator, Pedron et al additionally included those who were unemployed, homemakers, and those unemployed due to sickness in their “not retired” group. Similarly, Zheng included unemployed in their “not retired” group. This is an excerpt from our Discussion that raises this difference and how one study conducted a sensitivity analysis to address it:

“For example, the inclusion of those who are long-term unemployed in a “not retired” group may not make it a suitable comparator as long-term unemployment may be associated with chronic illnesses that have consequences on body weight; however, Pedron et al [51] showed that their results were statistically robust even with different definitions of their comparator.”

Comment #9

Results page 24: Suggestion for rephrasing the sentence: suggests that there may be a mediating role that sleep has in the impact of employment transitions on body weight change… to: suggests that sleep may play a mediating role in the relationship between employment transitions and body weight change…

Same page: you write: dietas outcomes. I have never come across this word before. Is it a typo or is it a real phrase?

We have updated the manuscript with the suggested rewrite of that sentence. The sentence now reads:

“Despite a lack of a formal analysis of mediation, Monsivais et al [12] suggests that sleep may play a mediating role in the relationship between employment transitions and body weight change for future research to consider.”

We have also corrected the typographical error identified from “dietas” to “diet as”.

Comment #10

Discussion page 26: you write: However, this linear rise in BMI may not be true for other countries. Perhaps you should mention here where the study is conducted, as this is not fresh in the readers memory.

We have updated the manuscript to specify the country that the Zheng study was conducted in. The sentence now reads:

“However, this linear rise in BMI may not be true for settings outside of the USA.”

Comment #11

Discussion page 27: you mention that operationalization of weight gain as a categorial or continuous variable has influenced the results in the studies by Zheng and by Forman-Hoffman et al. It seems that you only describe changes in BMI as a continuous variable and do not mention the categorical changes? I do not see from your discussion how categorical or continuous weight change has influenced the results.

Additionally, in the following section, you mention a study examining weight change as both categorical and continuous (Paige et al.); however, you only mention the results for the continuous variable.

I am a bit confused what you wish to discuss with these two paragraphs, and I do not think that the message is coming across very clear. Perhaps you should consider rewriting these paragraphs.

We thank the reviewer for identifying passages in our manuscript where we can improve the clarity of the writing. To clarify what we intended to convey, Zheng analyzed BMI change on a continuous scale while Foreman-Hoffman et al used a categorical outcome variable based on thresholds of 5% gain, 5% loss, or no change (between -5% and +5%). Both studies used a very similar sample from the Health and Retirement Study from the US.

In Zheng’s study, they had found men who retired from active occupations gained a statistically significant amount of weight; however, the actual gain is very small. In the Forman-Hoffman et al paper, they did not find that retired men have greater odds of weight gain of 5% of more compared to men who continued to work. We wished to summarize these two findings in a single sentence originally, but we acknowledge that it may have been written in an unclear manner originally: 

“as was the case with Zheng [14] and Forman-Hoffman et al [17] where men experienced increases in BMI but not beyond a threshold of a 5% change in BMI as their weight outcome”. 

With that sentence, we wished to highlight that differences in how outcomes are defined, will impact whether a statistically significant effect is detected or not, and even if one is detected, it may not be clinically meaningful. Neither paper operationalized their outcome in the other way, which is why we included some discussion based on a paper in another topic area that does try to capture change in a few different ways (Paige et al).

The Paige et al paper defined weight change on a continuous scale (kg) and as categories (1kg gain, 1kg loss, “no change”). Their finding on the continuous scale was that education was not associated with weight change. Their finding using multinomial logistic regression was that a higher educational attainment was associated with a lower likelihood of weight gain of 1kg and weight loss of 1kg (both ORs smaller than 1, with the “no change” group being the reference). This is what we wished to convey with this original sentence:

“The authors found no statistically significant or clinically meaningful change in body weight when the continuous variables were modeled [65]; however, they found that those with higher educational attainment had lower risk of both weight gain and loss of more than 1kg compared to those with no school certifications [65].”

To improve the clarity of our messaging, we have rewritten parts of these passages to be much more explicit in describing the definition of outcomes in both studies, and to minimize conflating results into single sentences.

“Operationalization of weight gain as a categorical or as a continuous variable, however, does seem to influence whether a statistically significant finding is detected, as was the case with Zheng [14] and Forman-Hoffman et al [17]. Change in BMI was measured on a continuous scale in Zheng [14]while change was categorized as 5% or greater gain, 5% or greater loss, or no change in Forman-Hoffman et al [17]. A statistically significant increase in BMI was found among retired men in Zheng [14], but the odds ratio for 5% or greater gain was not significant in Forman-Hoffman et al [17], suggesting men may gain weight following retirement that does not exceed clinical thresholds. Indeed, many of the findings of weight gain measured on a continuous scale are below clinically significant thresholds.”

---

## [Decision Letter · Decision Letter 1]

5 Aug 2022

A systematic review of evidence on employment transitions and weight change by gender in ageing populations

PONE-D-21-39118R1

Dear Dr. Tam,

We’re pleased to inform you that your manuscript has been judged scientifically suitable for publication and will be formally accepted for publication once it meets all outstanding technical requirements.

Kind regards,

George Vousden

Staff Editor

PLOS ONE

Additional Editor Comments (optional):

Reviewers' comments:

Reviewer's Responses to Questions

**Comments to the Author**

1. If the authors have adequately addressed your comments raised in a previous round of review and you feel that this manuscript is now acceptable for publication, you may indicate that here to bypass the “Comments to the Author” section, enter your conflict of interest statement in the “Confidential to Editor” section, and submit your "Accept" recommendation.

Reviewer #2: All comments have been addressed

2. Is the manuscript technically sound, and do the data support the conclusions?

Reviewer #2: Yes

3. Has the statistical analysis been performed appropriately and rigorously? 

Reviewer #2: N/A

4. Have the authors made all data underlying the findings in their manuscript fully available?

Reviewer #2: Yes

5. Is the manuscript presented in an intelligible fashion and written in standard English?

Reviewer #2: Yes

6. Review Comments to the Author

Reviewer #2: (No Response)

7. PLOS authors have the option to publish the peer review history of their article (what does this mean?). If published, this will include your full peer review and any attached files.

Reviewer #2: **Yes: **Maja Bramming

---

## [Editor Report · Acceptance letter]

10 Aug 2022

PONE-D-21-39118R1 

A systematic review of evidence on employment transitions and weight change by gender in ageing populations 

Dear Dr. Tam:

I'm pleased to inform you that your manuscript has been deemed suitable for publication in PLOS ONE. Congratulations! Your manuscript is now with our production department. 

Kind regards, 

on behalf of

Dr. George Vousden 

Staff Editor

PLOS ONE